# Mammary Gland Microbiota in Benign Breast Diseases

**DOI:** 10.3390/ijms26209951

**Published:** 2025-10-13

**Authors:** Nikita I. Ukraincev, Maria I. Kashutina, Larisa I. Kasatkina, Adkhamzhon B. Abduraimov, Yury V. Zhernov

**Affiliations:** 1Mammology Center, A.S. Loginov Moscow Clinical Scientific and Practical Center of the Moscow Healthcare Department, 111123 Moscow, Russia; 2Department of Public Health, National Research Centre for Therapy and Preventive Medicine of the Ministry of Health of the Russian Federation, 101990 Moscow, Russia; 3Department of Preventive Medicine, Reaviz Moscow Medical University, 117418 Moscow, Russia; 4A.N. Sysin Research Institute of Human Ecology and Environmental Hygiene, Centre for Strategic Planning and Management of Biomedical Health Risks of the Federal Medical and Biological Agency, 119121 Moscow, Russia; 5Department of General Hygiene, F. Erismann Institute of Public Health, I.M. Sechenov First Moscow State Medical University of the Ministry of Health of the Russian Federation (Sechenov University), 119435 Moscow, Russia; 6Fomin Clinic, 119192 Moscow, Russia; 7Center for Medical Anthropology, N.N. Miklukho-Maclay Institute of Ethnology and Anthropology of the Russian Academy of Sciences, 119017 Moscow, Russia

**Keywords:** breast disease, benign breast disease, breast tissue, mammary gland microbiota, dysbiosis, microbiome

## Abstract

The human microbiome is a critical factor in health and disease. While its association with breast cancer (BC) has been increasingly studied, this review provides a dedicated synthesis of the microbiota’s role in benign breast diseases (BBDs)—a common yet microbiologically overlooked spectrum of conditions. The primary aim of this work is to consolidate the current understanding of the composition, origins, and functional mechanisms of the mammary gland (MG) microbiota specifically in the context of BBD and to evaluate its potential for novel diagnostic and therapeutic targets. We detail the distinct MG microbiota, formed via exogenous (e.g., cutaneous, translocation) and endogenous (e.g., enteromammary, lymphohematogenous) pathways, and its interaction with the host through estrogen metabolism, immunomodulation, and epigenetic modifications. This narrative review reveals unique dysbiotic patterns in BBD, characterized by distinct microbial signatures, such as the enrichment of *Corynebacterium kroppenstedtii* in granulomatous mastitis and the presence of *Staphylococcus aureus* in fibroadenomas and lactational mastitis. Furthermore, specific gut microbial profiles are identified in BBD patients, including an increased abundance of genera such as *Clostridium* and *Faecalibacterium*, alongside a decrease in *Collinsella* and *Alistipes* compared to healthy controls. These specific taxa represent compelling candidates for diagnostic biomarkers. We conclude that microbial dysbiosis is a significant component of BBD pathogenesis. A paradigm shift toward multi-omics approaches and mechanistic studies is now essential to translate these associations into clinical applications. Understanding the BBD-specific microbiome holds the promise of revolutionizing patient care through microbiota-based diagnostics for differentiating benign subtypes and novel, personalized therapeutic strategies aimed at restoring microbial homeostasis.

## 1. Introduction

The human microbiome, comprising diverse microorganisms that exist predominantly as complex, structured communities known as biofilms, is a critical factor in maintaining health and influencing disease within the host’s internal environment and on its external surfaces [1,2]. These polymicrobial associations, composed of various types of microorganisms such as bacteria, viruses, archae, micromycete, and eukarya, colonize different human organs and tissues, forming what is known as the ‘microbiome’ [3]. The microbiome constitutes a complex ecosystem that can have a wide range of impacts on human health, which can be neutral, beneficial, such as vitamin synthesis and fiber degradation, or detrimental, such as infections and carcinogenesis [4,5,6]. While bacterial communities represent the most diverse and abundant component [7], the overall composition and stability of this ecosystem are crucial for host wellbeing.

The alteration in the composition of microbiota that occurs during dysbiosis frequently involves a reduction in microbial diversity, an increased prevalence of pathogenic microbes, and disruption of normal polymicrobial relationships [8]. This dysbiosis has been linked to the pathogenesis of various health problems, including acute and chronic inflammation, infections, allergies, and the development of tumors, both benign and malignant [8,9,10]. Consequently, scientific research has explored the role of microbes in tumor formation at various anatomical sites such as the intestines, lung, skin and others [11,12]. However, its role in breast pathology is only beginning to be understood.

Following this, the results of the Human Microbiome Project’s sequencing efforts revealed significant interindividual variability in microbial community diversity and abundance [4]. A multitude of factors are known to influence the formation of microbiomes and the incidence of dysbiosis [13], encompassing previous infections, inflammatory disorders [8], dietary habits, chemical exposures [14], and genetic predispositions [4,15].

Breast pathology is one of the most urgent public health problems, and breast cancer (BC), the most common type of cancer among women, is a significant socio-economic issue. According to the World Health Organization (WHO), 2.3 million women were diagnosed with breast cancer in 2022 [16]. The burden of this disease is substantial, profoundly impacting patients’ quality of life and posing high global costs [17,18].

In contrast, benign breast pathology, despite its high frequency, has received comparatively less scientific and clinical attention. Research shows that breast disease is mostly benign, although accurate data on its prevalence is not available [19,20]. In the presence of breast symptoms, breast cancer is detected only in 3–6% of cases. Consequently, there is a significant lack of evidence-based approaches for managing patients with benign breast conditions, since the main focus has been on optimizing diagnosis and treatment for breast cancer [21].

Notwithstanding these emerging associations, significant gaps persist in our mechanistic understanding of the breast microbiome’s role in pathogenesis. New studies suggest a link between certain microorganisms in breast tissue and the development of benign breast dysplasia. This finding is promising for the advancement of diagnostics and treatment methods. Critical unresolved questions include the precise taxonomic and functional definition of a ‘eubiotic’ breast microbiome; the causal mechanisms by which specific microbial consortia or their metabolites initiate or promote benign proliferative changes; the potential for microbial-host immune system crosstalk to modulate disease progression; and the translational applicability of microbial profiles as biomarkers for diagnostics, risk stratification, or novel therapeutic targets.

Research on the microbiome’s role in breast pathology is still in its early stages. Most studies focus on dysbiosis in breast tissues, such as the skin, nipple-areola complex secretions, ducts, and glandular fibrous tissues. Additionally, intestinal flora are also being investigated. This review aims to synthesize the current understanding of how human microbiomes might contribute to the development of benign breast diseases (BBDs), with a specific emphasis on identifying these fundamental knowledge gaps and outlining future research directions to address them.

## 2. Pathways of Mammary Gland Microbiota Formation and Mechanisms of Its Interaction with the Host Organism

Breast tissue obtained under aseptic conditions from patients without clinical signs of a local infectious process possesses its own unique microbiota [22,23,24]. A significant diversity of microorganisms is specific to breast tissue and distinct from the microbiota of the skin and other body sites [11,22,24,25]. Furthermore, the mammary gland (MG) is considered an organ with a low microbial biomass compared to other body areas [9].

### 2.1. Origin of the Mammary Gland Microbiota

The microbiota of breast tissue is formed through the influx of microorganisms via the nipple-areolar complex [23,26], translocation from the gastrointestinal tract and oral cavity [27], the urinary [28] and reproductive systems [22], the skin surface and other sources [29]. The unique MG microbiota is established via distinct exogenous and endogenous pathways, each characterized by specific mechanisms and influenced by key factors (Figure 1).

Exogenous sources refer to the introduction of microorganisms from the external environment. The cutaneous pathway is a primary route, whereby microbes from the skin surface translocate to the breast tissue. Although the microbiota of the MG and the overlying skin share species diversity, they remain distinct communities [24,29]. This pathway is significantly influenced by environmental factors (e.g., climate, geography, occupation, living conditions), lifestyle habits, particularly hygiene practices, and host physiology, such as a compromised skin barrier function. The evidence for this pathway is strong, primarily based on comparative 16S rRNA sequencing studies that show overlapping but distinct taxonomic profiles between skin and breast tissue samples [24,29].

Another critical exogenous route is the translocation pathway, which involves the retrograde inoculation of microbes via the lactiferous ducts [30,31,32,33,34,35]. This pathway is highly dependent on breastfeeding practices, including the type of feeding (direct breastfeeding, pumping, or formula), the mode of delivery (if breastfeeding), and the sex of the infant. Support for this pathway is moderate and indirect, largely inferred from studies detecting oral-associated bacteria in breast milk and the observed dynamic changes in milk microbiota in response to infant feeding patterns [32,35].

Medical interventions constitute two additional exogenous pathways. The artificial pathway describes the direct introduction of microbiota during procedures such as breast biopsies or surgery [36]. Conversely, the iatrogenic pathway refers to alterations in the existing microbiota induced by medical treatments, most notably the use of antibiotics or probiotics. Evidence for these exogenous pathways is currently limited and primarily hypothetical, based on analogous findings from other body sites, though this represents a critical area for future controlled studies.

Endogenous sources involve the translocation of microorganisms from within the host organism. The most well-studied is the enteromammary pathway, a sophisticated mechanism where gut and oral microbiota are transported to the MG via immune cells, such as CD18+ cells and dendritic cells [27,29,33,37,38,39,40]. This pathway creates a direct link between the gut and the breast, and its efficiency is influenced by the composition of the gut microbiota (e.g., states of dysbiosis associated with high body mass index (BMI)) and host diet.

The urogenital pathway proposes that microbiota from the urogenital tract may serve as a source for MG colonization [9], although the specific mechanisms are less defined. Furthermore, the local tissue environment, or intratissue pathway, dictated by the MG tissue structure (specifically the ratio of glandular/fibrous to adipose tissue) [41], shapes a unique ecological niche that determines which microbial communities can survive and thrive. Finally, the lymphohematogenous pathway involves the systemic dissemination of microbes to the breast through the lymphatic system and bloodstream [35], representing a potential route for bacteria from distant sites of infection or colonization.

In conclusion, the establishment of the mammary gland microbiota is a dynamic process governed by a multitude of exogenous and endogenous pathways. Understanding the intricate interplay between these routes and their modulating factors is crucial for elucidating the role of the microbiome in breast health and disease.

### 2.2. Microbiota of Unaltered Mammary Gland Tissues

The composition of the microbial community in unaltered mammary gland tissues has been characterized through diversity analyses. These analyses reveal that while breast tissue harbors a greater number of bacterial species (alpha-diversity) than skin, the communities are distinct in their composition (beta-diversity), primarily due to differences in rare or less abundant species [24,42]. The resulting foundational profile is dominated by the species listed in Table 1 and Figure 1.

The higher prevalence of the phyla Pseudomonadota (Proteobacteria) and Bacillota (Firmicutes) compared to other taxonomic groups may be due to the affinity of these microorganisms for the fatty acid-rich environment of breast tissues [4]. Disruption of polymicrobial interactions leading to dysbiosis may contribute to the development of breast diseases [43].

**Table 1 ijms-26-09951-t001:** Representatives of the bacterial community identified in unaltered female breast tissue.

Phylum	Family	Genus	Species	References
Pseudomonadota (Proteobacteria)	Sphingomonadaceae	-	-	[44]
	Methylobacteriaceae	*Methylobacterium*	-	[4,9]
	Burkholderiaceae	*Ralstonia*	-	[4,44,45]
	Sphingomonadaceae	*Sphingomonas*	*yanoikuyae*	[4]
			-	[22,24,44]
	Pseudomonadaceae	*Pseudomonas*	*-*	[22,44,46]
	Comamonadaceae	*-*	-	[22]
	Enterobacteriaceae	*-*	-	[22]
	Moraxellaceae	*Acinetobacter*	-	[22]
	Pasteurellaceae	*Haemophilus*	-	[46]
	Neisseriaceae	*Neisseria*	-	[46]
	-	*-*	-	[4,9,22,26,45]
Bacillota (Firmicutes)	Veillonellaceae	*Veillonella*	-	[46]
	Staphylococcaceae	*Staphylococcus*	-	[22,44]
	Streptococcaceae	*Lactococcus*	-	[44]
		*Streptococcus*	-	[44]
	Clostridiaceae	*Clostridium*	-	[44]
	Lactobacillaceae	*Lactobacillus*	-	[44]
	Listeriaceae	*Listeria*	*welshimeri*	[22]
	-	-	-	[4,9,22,26,45,47]
Actinomycetota (Actinobacteria)	Corynebacteriaceae	*Corynebacterium*	-	[44]
	Micrococcaceae	*Micrococcus*	-	[4]
	Propionibacteriaceae	*Propionibacterium*	-	[22,44]
	-	*-*	-	[4,26]
Bacteroidota (Bacteroidetes)	Bacteroidaceae	*Bacteroides*	-	[9]
	Prevotellaceae	*Prevotella*	-	[22,44]
	-	*-*	-	[9,26,44]

In recent years, several studies have clarified the role of viruses and fungi (micromycetes) in the human microbiome [48]. For instance, DNA from human papillomavirus (HPV), Epstein–Barr virus (EBV), human cytomegalovirus (HCMV), herpes simplex virus (HSV), and human herpesvirus type 8 (HHV-8) has been detected in breast tissue both under normal conditions and with pathological changes [49,50]. Given that viruses are obligate parasites, it is difficult to distinguish a separate infectious process in the MG from long-term viral persistence in breast tissue cells with potential carcinogenicity—in this case, one can refer to the MG virome. The predominant focus on the bacterial fraction of the human microbiome in most studies has led to an incomplete understanding of the role of micromycetes in polymicrobial interactions. Research into the role of fungal communities (the mycobiome) is complicated by the low fungal biomass in breast tissues compared to the bacterial component. According to published studies, *Candida albicans* and *Saccharomyces* are identified in normal breast tissue structure, while *Malassezia*, *Davidiella*, *Sistotrema*, and *Penicillium* are found in breast milk during lactation [6,30]. However, their specific role of viruses and fungi in the pathogenesis of BBD remains largely speculative.

### 2.3. The Microbiota-Human Organism Interaction

The MG microbiota interacts with the host organism by regulating immune, metabolic, transcriptional, and epigenetic processes through the production of enzymes and other biologically active substances [4,35,41,51,52,53,54].

Bacterial metabolites (e.g., short-chain fatty acids, acetate, butyrate, pyruvate, formate, amines (cadaverine), bile acid derivatives (lithocholic acid, deoxycholic acid), indole, etc.) can influence processes of cell growth, apoptosis, epithelial–mesenchymal transition, anti-tumor immunological surveillance, and also exert cytotoxic and genotoxic effects [55,56]. Dysbiosis leads to changes in the bacterial metabolite profile [56,57]. Some bacterial metabolites and their mediated effects are presented in Table 2.

Furthermore, microorganisms inhabiting remote areas of the body can influence each other through metabolites and immune factors [12,40]. These interactions can impact the onset and development of pathological conditions, particularly breast diseases [58,59,60].

### 2.4. Mechanisms of Microbiota-Induced Breast Pathogenesis

Several hypotheses have been proposed regarding the ability of microorganisms to cause DNA damage, modulate estrogen metabolism, shape the immune microenvironment of the breast—inducing chronic inflammation and local immunosuppression—and potentiate proliferative processes (Figure 2) [11,61,62,63].

One of the primary mechanisms involves the regulation of estrogen metabolism. Elevated estrogen levels are closely associated with the activation of estrogen receptors, promoting proliferative processes in breast tissue in BBD and BC [11,64,65]. The gut microbiota regulates processes of enterohepatic circulation of active estrogen forms [66,67]. and the synthesis of estrogen-like substances [57,60,61]. The collection of microorganisms influencing systemic levels of estrogen and its metabolites constitutes the concept of the estrobolome [68,69]. The role of local estrogen levels in breast tissue in BC should also be noted [9].

Beyond hormonal influence, the microbiota exerts a significant effect through local immunomodulation and the maintenance of chronic inflammation. Available research data indicate that commensal microflora of the gastrointestinal tract possesses immunomodulatory properties. The gut microbiota can maintain chronic inflammation by altering the balance between proliferation and apoptosis and triggering unregulated innate and adaptive immune responses. Immunoglobulin A (IgA), which maintains mucosal barrier integrity, is involved in recognizing and regulating the composition of the gut microbiota [23,57,70].

A further mechanism by which the microbiota may influence breast tissue is through epigenetic modifications. Previous studies have shown that the microbiota can influence processes of epigenetic regulation [71]. Epigenetic mechanisms include DNA methylation, acetylation and deacetylation of histone proteins, and modification of non-coding RNAs and microRNAs. Epigenetic modifications lead to enhanced or attenuated cell growth and regulation of cellular signaling pathways. Therefore, studying epigenetics as one of the mechanisms by which the microbiota influences pathological states of the breast appears promising [40].

## 3. Microbiota Composition and Diversity in BBD

### 3.1. General Characteristics of BBD

BBD is among the most common breast diseases and includes conditions varying by clinical, morphological, and etiological criteria. These conditions are based on an imbalance between the epithelial and connective tissue components due to specific features of proliferation and regression processes in breast tissue. Given the heterogeneity of benign pathology, a number of terms are used to denote it: BBD, mastopathy, fibrocystic mastopathy, fibrocystic changes, fibrocystic disease, dyshormonal hyperplasia of the mammary glands, fibroadenomatosis, etc. The prevalence of benign breast changes in the female population is 50% or higher. Considering the prevalence and the increased risk of developing BC with certain types of benign changes, the issues of improving diagnosis, differential diagnosis, and treatment are relevant [72,73].

BBD is most often classified by clinical manifestations, degree of proliferative changes, type of pathomorphological changes on biopsy, etc. (Figure 3 and Figure 4) [21,72,74,75,76,77]. Breast tissue changes on histological examination are classified from B1 to B5 depending on the degree of suspicion for BC [21,78].

A special group of diseases consists of lesions of uncertain malignant potential (B3). Some proliferative breast conditions may be associated with the presence of DCIS, invasive BC, or be markers of an increased risk of developing BC [57,79,80]. Such conditions are classified as lesions of uncertain malignant potential (B3) on histological examination [21]. These include atypical ductal hyperplasia (ADH), flat epithelial atypia (FEA), classical lobular neoplasia (LN, ALH, LCIS), as well as lesions of heterogeneous structure with a risk of incomplete sampling during biopsy—intraductal papilloma with/without atypia, radial scar, and complex sclerosing lesion [21].

### 3.2. General Characteristics of Microbiota in BBD

The main task of clinical mammology is the differential diagnosis of benign and malignant breast neoplasms. Consequently, research is primarily aimed at identifying the association between changes in the microbiota of breast tissues and gut flora in BC, while studies assessing the features of the microbiota in BBD have been published in limited numbers to date. Most of them are aimed precisely at comparing and identifying similarities and differences in the tissue microbiota of unaltered breast tissues and tumor formations (if present) and gut microflora in the absence of breast pathology, BC, and BBD [24,43,60].

Studies of taxonomic profiles show that the bacterial microbiota of breast tissue in BC and BBD is similar, dominated by representatives of the phyla Bacteroidota (Bacteroidetes), Bacillota (Firmicutes), Pseudomonadota (Proteobacteria), and the phylum Actinomycetota (Actinobacteria) [11,24,43]. However, more granular analysis reveals taxon-specific differences that may hold diagnostic potential. In unaltered adjacent tissues in BC, a significantly greater abundance of certain bacterial taxa was found, including the genus *Bacillus*, *Staphylococcus*, and members of the family Enterobacteriaceae, compared to patients with benign breast tumors and healthy women [43]. Moreover, the relative abundance of Pseudomonadota (Proteobacteria) in benign diseases was found to be significantly lower than in malignant ones [11]. While these findings are descriptively valuable, their utility as standalone diagnostic biomarkers is currently limited by small cohort sizes and methodological variability. The critical question of whether these microbial signatures are drivers of pathogenesis or merely secondary consequences of the altered tumor microenvironment remains unresolved.

In BBD, a low level of bacteria influencing DNA damage processes is noted, which may be a possible factor preventing malignant transformation [43]. This observation suggests a potential functional role for the microbiota in disease progression beyond mere taxonomic composition. Taxonomic profiles in benign tumors are more similar to profiles of normal adjacent tissue in women with malignant tumors than to profiles of tissues from healthy patients [43]. This similarity could indicate a shared microenvironmental niche or an early, ‘field effect’ change that predisposes tissue to neoplasia, warranting investigation into whether specific microbial consortia can predict the risk of malignant transformation in BBD. Beta-diversity assessment shows that the microbial community in unaltered breast tissue adjacent to invasive cancer differs from that in women with benign diseases, mainly due to rare and/or less abundant species. Alpha-diversity analysis revealed no significant differences [4,24]. The diagnostic value of these beta-diversity shifts, particularly the role of low-abundance taxa, is a promising but unvalidated area for future research aimed at developing non-invasive microbial biomarkers for differential diagnosis.

Metabolic pathways mediated by the tissue microbiota also differ between BC and BBD. According to KEGG (Kyoto Encyclopedia of Genes and Genomes) PATHWAY, benign tissues exhibit elevated metabolism of cysteine and methionine, glycosyltransferases, and fatty acid biosynthesis, while the microbiota of malignant tissues demonstrates reduced inositol phosphate metabolism [24]. Bacterial lipopolysaccharides are often present in many human solid tumors [21,60].

A number of viruses possess oncogenic potential by initiating malignant transformation of epithelial cells, prolonging the cell cycle, activating cell proliferation, and preventing apoptosis [35]. The phenomenon of the influence of androgens and estrogens on the replicative activity of some viruses makes viruses a possible etiological factor in the development of a number of hormone-sensitive tumors and conditions, particularly of the breast.

For example, HPV can integrate into the cell genome and mediate oncogenic transformation via E6 and E7 proteins. HPV DNA was identified in 20.3% of malignant neoplasms and 35% of benign neoplasms of the breast [49]. It was also detected in 27.3% of biopsy samples of unaltered breast tissue. It was not found in biopsy material of in situ neoplasms or borderline lesions. HCMV can modulate the tumor immune microenvironment and stimulate tumor cell growth. In breast tissues, HCMV genetic material is more frequently detected in malignant neoplasms than in normal tissue. Very low levels of EBV DNA are detected in breast tissues in BC by quantitative PCR. Thus, the role of EBV and HCMV in the development of breast pathology, primarily breast cancer, remains debatable [21].

Colonization of epithelial tissues and cells by some species (tropism) of micromycetes may indicate a potential role of fungi in breast diseases [51]. Available studies only highlight the association of micromycetes with malignant neoplasms. Thus, micromycetes have been associated with 35 cancer types, yet their role in carcinogenesis remains unexplored [48,51]. In the context of breast cancer (BC), the most significant fungal-bacterial-immune interactions have been identified, particularly involving representatives of the genera *Aspergillus*, *Malassezia*, and *Cladosporium* (*Cladosporium sphaerospermum*) [81]. By analogy, it is plausible that similar polymicrobial crosstalk could occur in the inflammatory milieu of BBD, such as duct ectasia or mastitis, but this remains a hypothesis awaiting clinical validation.

Fungi remain understudied but important commensals/opportunistic pathogens that shape unique host immune responses. Research has demonstrated that specific fungi like *Malassezia* contribute to oncogenesis through mechanisms including pro-inflammatory cytokine induction and complement activation, collectively establishing an immunosuppressive tumor niche [81]. By extension, it is plausible that fungal colonization of the breast ductal system may similarly influence local immunodynamics in BBD. For instance, persistent fungal antigens could potentially sustain granulomatous inflammation in conditions such as idiopathic granulomatous mastitis. Given the possibility of symbiotic and antagonistic interactions (physical, biochemical) between bacteria and fungi, further research is required to assess the role of fungi in the polymicrobial interaction of the tumor environment. If a distinct mycobiota is associated with BBD, it could open new avenues for management, such as the use of antifungal or immunomodulatory therapies in refractory cases, similar to approaches used in other fungal-driven inflammatory conditions. Defining the breast mycobiota could ultimately help identify patients with specific microbiological risk profiles for disease progression, but this potential remains entirely speculative without foundational data.

### 3.3. Features of Gut Microbiota in Breast Pathology

The role of gut dysbiosis in the development of BBD is also currently debatable. In patients with malignant breast tumors, the species diversity of the gut microbiota is lower but more homogeneous than in patients with benign tumors [59]. It should be noted that in a number of studies on the composition of gut microbiota in BBD and BC, no statistical differences were observed in the assessment of α- and β-diversity [60].

When comparing patients with BBD and healthy individuals, no differences in α-diversity indices were found [57].

Furthermore, assessment by beta-diversity noticeably differed among the three groups (BC/BBD/normal). These results indicate an altered composition of the gut microbiota in healthy women, BC patients, and BBD patients [57].

An increase in the number of representatives of the genera *Clostridium*, *Faecalibacterium*, *Lachnospira*, *Romboutsia*, *Fusicatenibacter*, *Xylophilus*, *Arcanobacterium*, *Escherichia*, *Peptoniphilus*, *Coprobacillus*, *Lactobacillus*, *Porphyromonas*, and the family Erysipelotrichaceae in the gut microbiota of patients with benign tumors has been reported, along with a decrease in the abundance of the genera *Collinsella*, *Alistipes*, *Megamonas*, *Butyricimonas*, *Acidaminococcus*, *Asaccharobacter*, *Tissierella*, and *Cloacibacillus* [57,58,60].

The metabolic pathways of the gut microbiota in patients with malignant tumors differed significantly from those in patients with benign neoplasms [60]. In patients with malignant tumors, unlike BBD, marked activation of lipopolysaccharide biosynthesis pathways was noted. Conversely, KEGG analysis revealed significant activation of sporulation in patients with benign tumors [60,82].

### 3.4. Mammary Gland Microbiota in Specific Types of BBD

Current scientific data do not cover the entire spectrum of microbiota changes across all types of BBD. Research is focused mainly, as noted earlier, on fundamental differences in microbiota (gut and breast tissue) in malignant and benign breast conditions in general. Below, a number of specific benign breast pathologies and their association with the composition of tissue and gut microbiota are considered. Summary data on associated and protective microorganisms for various BBDs are presented in Table 3 and Figure 1.

#### 3.4.1. Breast Cysts

Among representatives of the gut microbiota, the family Alcaligenaceae is associated with an increased risk of developing breast cysts, while the genera *Eubacterium ruminantium* and *Lactococcus* are associated with a reduced risk [23,57]. Cysts can be complicated by secondary infection with clinical and radiological signs of a complicated cyst (cyst with inflammation); see the Section 3.4.7.

HPV was detected in 40.0% of biopsies taken from patients with fibrocystic mastopathy [83].

#### 3.4.2. Breast Fibroadenomas

*Staphylococcus aureus* is an important factor causing mutation of the MED12 gene, which may contribute to the development of breast fibroadenoma and uterine leiomyoma [57,84]. Among benign neoplasms, HPV DNA was identified in 38.9% of histological material from fibroadenomas [49].

#### 3.4.3. Lactational Mastitis

Lactational mastitis is the most common type of mastitis, accounting for 33% of all breast diseases [85]. In 90% of cases, the etiological agent is Staphylococcus aureus, less frequently coagulase-negative *Staphylococcus*, *Streptococcus*, *Pseudomonas aeruginosa*, and *Escherichia coli* [21]. The presence of representatives of the genera *Anaerofilum* and *Anaerotruncus* in the gut microbiota is associated with cases of lactational mastitis, while the genus *Butyricimonas* and the orders *Coriobacteriales*, *Pasteurellales*, and *Verrucomicrobiales* had a negative association [23]. In this regard, probiotics are used in the treatment of lactational mastitis and are potentially effective for chronic and subclinical forms of mastitis as an alternative to antibacterial therapy [57].

#### 3.4.4. Non-Lactational Mastitis

Non-lactational mastitis accounts for up to 3% of all benign breast diseases [85]. Given the clinical picture and difficulties in diagnosis and treatment, non-lactational mastitis can cause distress in some women. Non-lactational mastitis occurs at any age, although it more often affects young and middle-aged women [85]. The pathological forms of this disease are diverse, including duct ectasia (MDE), periductal mastitis (PDM), and granulomatous lobular mastitis (GLM) [85].

The influence of the MG microbiota on the development of inflammatory breast diseases has been studied [29,86,87]. Representatives of the gut flora, for example, the family Prevotellaceae, are associated with inflammatory breast changes [23]. The breast tissue microbiota in patients with non-lactational mastitis demonstrates differences in the composition of microbial communities compared to healthy patients. In patients with non-lactational mastitis, representatives of bacterial communities inhabiting the gut, particularly the genera *Ruminococcus*, *Coprococcus*, and *Clostridium*, were identified in breast tissue [85]. Non-lactational mastitis can also be associated with autoimmune reactions [85].

#### 3.4.5. Granulomatous Mastitis

Granulomatous mastitis is a rare inflammatory breast disease in women of reproductive age [29,88]. The etiology of this condition is unknown; nevertheless, a significant role is assigned to local dyshormonal changes, hyperprolactinemia, autoimmune reactions, and infectious agents [21,85]. A role for the genus *Corynebacterium*, particularly the species *Corynebacterium kroppenstedtii*, has been identified in the pathogenesis of granulomatous inflammation [85,87]. Other taxa encountered in granulomatous mastitis include representatives of the genera *Pseudomonas*, *Brevundimonas*, *Stenotrophomonas*, *Acinetobacter*, and *Aspergillus* [29].

#### 3.4.6. Ductal Changes

Periductal mastitis is an inflammatory disease of the subareolar lactiferous ducts, with a prevalence of up to 9% outside the lactation period [89]. In nipple discharge, bacterial flora is detected in 50% of cases, while against the background of duct ectasia, it is detected in 62% of cases [85]. This may be associated with structural changes in the duct wall during a persistent inflammatory process against the background of combined infections caused by representatives of the genera *Enterococcus*, *Streptococcus*, and *Bacteroides* [85,90].

#### 3.4.7. Purulent-Septic Changes of the Breast

In abscesses associated with non-lactational mastitis, the most frequent bacterial strains were coagulase-negative staphylococci and peptostreptococci, *Staphylococcus aureus* (50% of cases MRSA) [91], including as part of combined bacterial infection [21,85]. Certain diseases associated with impaired skin barrier function, such as atopic dermatitis, promote contamination of deep skin layers and underlying tissues, causing the development of an infectious process, including breast abscesses [29]. Representatives of the genera *Corynebacterium* and *Pseudomonas* (*Pseudomonas aeruginosa*) are often associated with infections of the skin and underlying soft tissues (abscesses, phlegmons, fistulas, etc.) [92,93]. These skin microorganisms are capable of metabolizing fatty acids and are considered potential pathobionts in breast tissues [29].

#### 3.4.8. Fibrous Capsular Contracture

The transfer of microorganisms during surgery via surgical instruments can lead to opportunistic subclinical infection. Staphylococci, *Cutibacterium acnes*, *Pseudomonas aeruginosa*, Staphylococcus lugdunensis, Staphylococcus hominis, *Staphylococcus epidermidis*, *Sphingomonas paucimobilis*, and *Aeromonas salmonicida*, which are found on the skin of healthy individuals, are also frequently identified in infections associated with surgical interventions [36,94,95]. The frequency of infectious-inflammatory complications in breast surgeries ranges from 3 to 15%. Despite prophylactic antibiotic use and adherence to aseptic and antiseptic principles, positive culture results (breast tissue) were found in 20.4% of cases [96].

Augmentation mammoplasty is a common breast surgery and is associated with the development of capsular contracture [97]. Up to 56% of capsular contracture cases are associated with the detection of bacterial flora on the surface of implants or in the fibrous capsule, particularly *Staphylococcus epidermidis* [28,98]. Furthermore, species isolated from the structure of the fibrous capsule included *Escherichia coli*, *Diaphorobacter nitroreducens*, *Cutibacterium acnes*, *Staphylococcus aureus*, and *Staphylococcus* spp. [97,99]. The formation of bacterial biofilms on the implant surface promotes resistance to antibiotic therapy and allows microorganisms to escape immune surveillance, also complicating the assessment of the species composition of the bacterial flora [97].

#### 3.4.9. Anaplastic Large Cell Lymphoma

In addition to capsular contracture, bacterial colonization and biofilm formation are linked to breast implant-associated anaplastic large cell lymphoma (BIA-ALCL) [100]. The species *Staphylococcus saprophyticus* and representatives of the genus *Ralstonia* are the most frequently detected microorganisms in BIA-ALCL, both on the side of interest and in the contralateral breast [97,101].

## 4. Breast Microbiota in Men

Before puberty, breast tissue is identical in both sexes. During puberty, boys experience transient proliferation of the milk ducts and stroma (due to estrogen stimulation), followed by involution of these structures (with increasing testosterone levels) [46,102]. The male breast does not develop terminal ductal lobular units due to the absence of progesterone [102]. The spectrum of pathology in the male breast is limited and related to the gender-specific histological structure of the organ [46].

Most hyperplastic processes in the male breast are benign, etiologically and pathogenetically similar to changes in the female breast. Male breast cancer (1 per 100,000 in Europe with a peak at 71 years) is rare, accounting for approximately 1% of breast pathologies [46,102].

Benign Conditions of the Male Breast [46,102]:Developmental anomalies (amastia, polymastia, nipple inversion, athelia, polythelia, etc.);Inflammatory and reactive changes (mastitis, abscess, Mondor’s disease, etc.);Ductal changes (duct ectasia, intraductal papilloma, etc.);Systemic diseases/symptoms of systemic diseases (diabetic mastopathy, gynecomastia);Benign neoplasms (lipoma, angiolipoma, cavernous hemangioma, myofibroblastoma, epidermal cysts, pseudoangiomatous stromal hyperplasia (PASH), hamartoma, etc.);Traumatic and post-traumatic changes (hematoma, fat necrosis).

Gynecomastia is the most common pathological benign condition of the male breast. The prevalence of gynecomastia is high, especially in the neonatal period (60–90%), during puberty (48–64%), in the reproductive period (up to 30%), and in the elderly (60% after 70 years). The development of gynecomastia is often associated with transient physiological changes, endocrine disorders, systemic diseases, drug therapy, and can also develop idiopathically [103]. The development of glandular tissue creates a morphological substrate for the development of other breast pathologies.

A critical barrier in this field is the paucity of data concerning the microbiota of the male breast. The existing literature is exceptionally limited and precludes definitive conclusions.

A critical barrier in this field is the paucity of data concerning the microbiota of the male breast. The existing literature is exceptionally limited and precludes definitive conclusions. While the concept of a sexually dimorphic breast microbiome—a microgenderome—has been proposed for both cancerous and histologically normal tissues, the evidence base for male-specific characterization is nascent. A single study has reported that histologically normal male breast tissue exhibits greater microbial richness (alpha-diversity) and distinct community composition (beta-diversity) compared to female tissue, noting a predominance of Bacteroidaceae, Caulobacteraceae, Comamonadaceae, Enterococcaceae, Microbacteriaceae, Peptoniphilaceae families, and the genera *Brevundimonas*, *Clavibacter*, *Comamonas*, and *Rhodococcus* [46]. However, these findings are derived from a limited sample set and must be considered preliminary.

Consequently, there exists a substantial research gap in the comprehensive characterization of the male breast microbiome and its potential role in health and disease states, such as gynecomastia and male breast cancer. Present clinical paradigms for diagnosing and managing breast pathologies in men remain largely extrapolated from studies conducted in female populations. Therefore, a targeted investigation of the male breast microbiota is warranted to elucidate its pathophysiological significance and to determine whether microbial dysbiosis contributes to disease etiology in a sex-specific manner.

## 5. Limitations in Studying the Microbiota in BBD

This narrative review of available scientific data on the microbiota in BBD reveals a field in its early stages, characterized by several significant challenges. These limitations can be broadly categorized into methodological and analytical, clinical, and conceptual hurdles that must be addressed to advance the field.

### 5.1. Methodological and Analytical Limitations

The technical study of the breast microbiome presents unique difficulties. Foremost among these is the inherently low microbial biomass of breast tissue, which amplifies the risk of contamination during sampling and sequencing and complicates data analysis and interpretation [21]. This issue is exacerbated by a lack of standardized, cross-sectional studies and unified protocols. Existing research often involves small, heterogeneous patient cohorts and employs varied methods for DNA extraction, sequencing, and bioinformatic analysis, making meta-analysis and direct comparison of findings across studies unreliable [31]. Furthermore, the current characterization of the breast microbiome is incomplete, with a pronounced lack of high-quality data on the virome and mycobiome, focusing almost exclusively on bacterial communities.

### 5.2. Clinical and Cohort-Related Limitations

From a clinical perspective, obtaining appropriate samples for research is a major obstacle. There is a critical paucity of baseline data on the ‘healthy’ breast microbiome due to the obvious inaccessibility of truly normal breast tissue from healthy individuals. Consequently, studies often rely on adjacent histologically normal tissue from cancer patients as controls, which may not represent a genuine healthy state and could introduce bias. This is compounded by a dominant research focus on malignant breast neoplasms, which has left the BBD microbiota comparatively neglected. The diagnostic category of ‘lesions of uncertain malignant potential (B3)’ introduces additional complexity, as the ambiguous nature of these lesions makes it difficult to define clean and distinct patient cohorts for microbiological studies.

### 5.3. Conceptual and Interpretative Limitations

Beyond technical and clinical issues, there are profound conceptual challenges. The most significant is the inability to infer causality from current data. It remains unknown whether observed microbial dysbiosis is a causative factor in the pathogenesis of BBD, a consequence of the altered tissue microenvironment, or merely a bystander effect [46]. This is complicated by the high degree of inter-individual variability in the microbiota and the multifactorial etiology of BBD, making it difficult to disentangle the specific contribution of microbes from other genetic, hormonal, and environmental factors. Finally, while the potential for microbial signatures to serve as clinical biomarkers is a compelling goal, the transition from correlative observations to validated, actionable diagnostic or therapeutic tools for personalized medicine remains a considerable challenge [60,104].

In conclusion, addressing these multifaceted limitations is paramount for building a robust scientific foundation. Future research must prioritize standardized protocols, larger prospective cohorts, and innovative analytical methods to move from descriptive association to mechanistic understanding, ultimately unlocking the potential of the microbiome for the prevention, diagnosis, and treatment of BBD.

## 6. Materials and Methods

This narrative review was conducted by systematically searching the PubMed, Scopus, and Google Scholar databases for literature published between January 2000 and June 2025. The search utilized a comprehensive set of keywords and their combinations, including ‘breast microbiota’, ‘mammary gland microbiota’, ‘breast tumor microbiota’, ‘benign breast diseases’, ‘benign breast lesions’, ‘breast tissue microbiota’, and ‘human mammary gland microbiota’. Duplicate records were removed, and the search was restricted to titles and abstracts focusing on microbiological findings in human breast tissue, milk, or associated bodily fluids (e.g., urine, nipple aspirate fluid) in the context of both benign and malignant breast conditions to provide a comparative context. The initial search yielded a total of 2815 studies. After screening titles and abstracts for relevance to the mammary gland microbiota and benign breast diseases, 215 full-text articles were assessed for eligibility. Studies were excluded if they focused solely on cancer without a benign disease comparison, were purely methodological, or involved animal models. Ultimately, a final set of 85 studies was included in the synthesis. While not a systematic review, this approach aimed to comprehensively cover the current state of knowledge on the features of the microbiome in patients with benign breast pathology.

## 7. Conclusions

Accumulating evidence underscores a compelling association between microbial dysbiosis and the pathogenesis of breast diseases. This review synthesizes current understanding of the MG microbiota—a distinct community formed through exogenous (cutaneous, retrograde translocation) and endogenous (enteromammary, hematogenous) pathways—and its intricate interactions with the host via regulation of estrogen metabolism, immunomodulation, and epigenetic mechanisms.

A central and unresolved question is whether the observed microbial shifts in BBD are a cause or a consequence of the disease. While the low microbial biomass and correlative nature of most studies preclude definitive conclusions, several lines of evidence suggest a potential causative role for specific taxa. For instance, the consistent association of *Corynebacterium kroppenstedtii* with the lipid-rich microenvironment of granulomatous mastitis and the established pathogenicity of *Staphylococcus aureus* in acute lactational mastitis point to microbes that may actively drive pathology.

Looking ahead, the translation of microbiota research into clinical practice for BBD represents a key frontier. Within the next 5–10 years, we anticipate that validated microbial signatures could augment traditional diagnostics. Specifically, the relative abundance of Pseudomonadota or the presence of a *Corynebacterium kroppenstedtii*-dominant profile in biopsy samples or ductal lavage fluid could serve as a molecular tool to differentiate inflammatory BBD subtypes or stratify the risk of malignant transformation in proliferative benign conditions. This could lead to the development of non-invasive microbial risk scores to guide patient management.

To realize this clinical potential and move beyond correlation to causation, future research on BBD must be unequivocally directed toward a multi-omics and mechanistic framework. The critical next steps should include longitudinal cohort studies specifically tracking microbial dynamics across different BBD subtypes—from initial development through potential progression or resolution. This should be coupled with integrated meta-omics approaches to define functional relationships between microbial communities and BBD pathogenesis, particularly focusing on inflammatory processes in conditions like granulomatous mastitis and proliferative changes in benign tumors. Furthermore, developing BBD-specific experimental models, including patient-derived organoids from fibroadenoma and duct ectasia tissues, will be crucial for validating the functional role of candidate microbes and testing potential microbiota-modulating therapies.

Ultimately, advancing this field necessitates strengthened interdisciplinary collaboration across microbiology, oncology, and immunology to translate these fundamental insights into tangible benefits for patients with benign breast diseases.

## Figures and Tables

**Figure 1 ijms-26-09951-f001:**
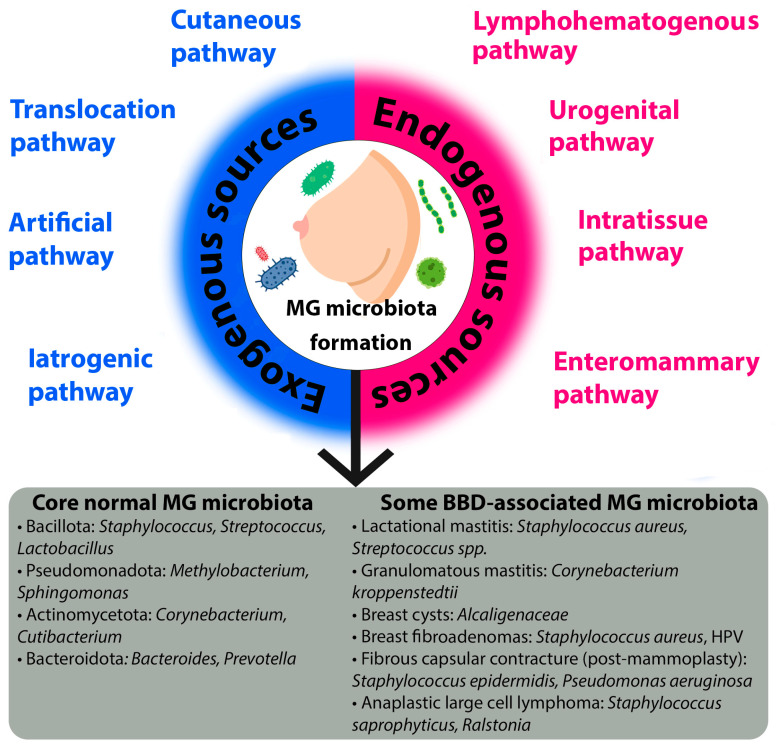
Exogenous and endogenous sources of mammary gland microbiota formation (MG—mammary gland; BBD—benign breast disease; HPV—human papillomavirus).

**Figure 2 ijms-26-09951-f002:**
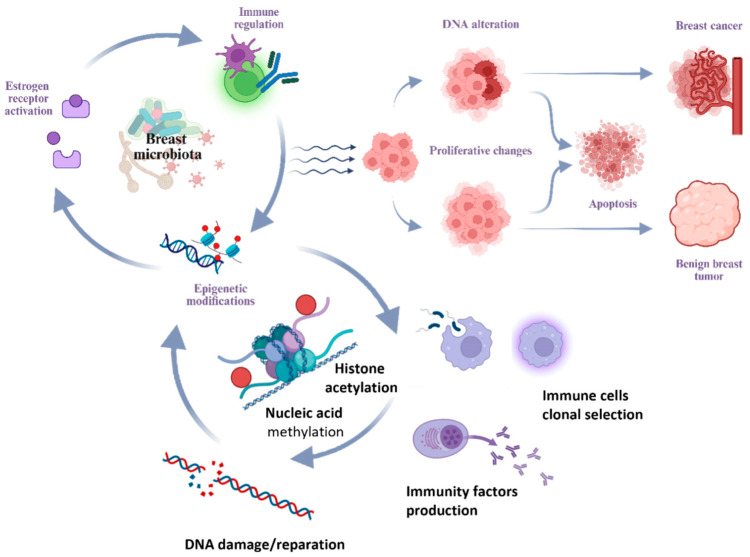
Main mechanisms of microbiota influence on the development of breast pathology.

**Figure 3 ijms-26-09951-f003:**
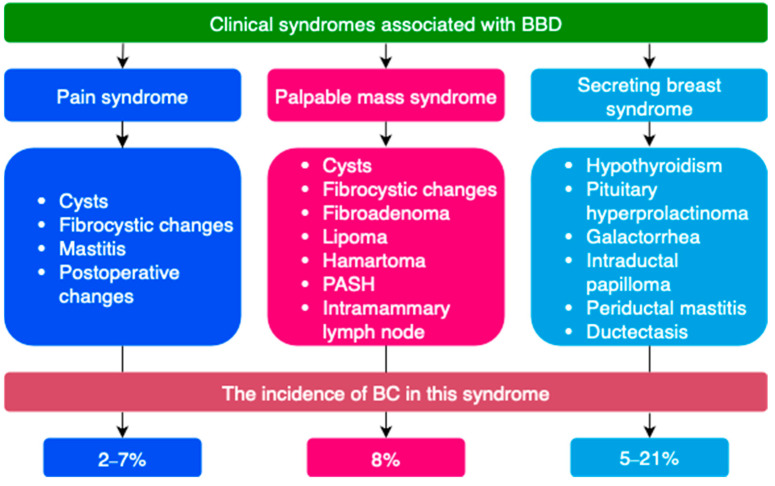
Classification of benign breast tissue changes by dominant clinical syndrome (Adapted with permission from Stachs A. et al. [21]. Copyright 2019, Deutscher Ärzteverlag GmbH).

**Figure 4 ijms-26-09951-f004:**
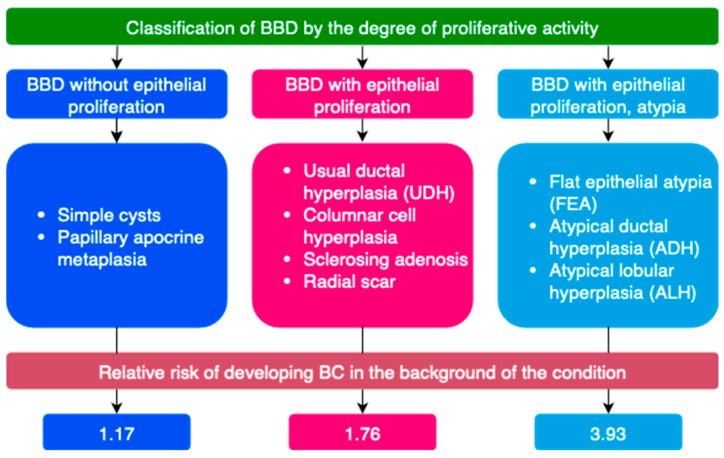
Classification of benign breast tissue changes by degree of proliferative activity (Adapted with permission from Stachs A., et al. [21]. Copyright 2019, Deutscher Ärzteverlag GmbH).

**Table 2 ijms-26-09951-t002:** Some significant bacterial metabolites and their mediated metabolic effects.

Bacterial Metabolite	Metabolic Effect	References
Cytolethal distending toxin (CDT) and colibactin	Promotes DNA double strand breaks (DSB)	[35]
Rho GTPase family proteins	Reorganizing actin cytoskeleton	[35]
Cadaverine	Endothelial to mesenchymal transition modulation	[4]
Lithocholic acid (LCA)	Increases oxidative stress. Regulates KEAP1, NRF2, TGR5, GPX3 expression	[4]
Lipopolysaccharides (LPS)	Associated with S100A7 expression—regulates mammary cell proliferation	[4]
Trimethylamine N-oxide (TMAO)	Effects cell proliferation by α-casein	[4]
β-glucuronidase and/or β-glucosidase	Promote recirculation of estrogen and estrogen-like metabolites	[31]
Short chain fatty acids, folates, biotin	Activate epigenetically silenced genes in cells such as p21, BAK etc.	[13,31]

**Table 3 ijms-26-09951-t003:** Microbiota in benign female breast diseases.

Disease/Condition	Microorganisms/Taxa (Associated)	Microorganisms/Taxa (Protective/Risk-Reducing)	Location/Notes	References
Breast cysts	Family Alcaligenaceae (gut)	Genus *Eubacterium ruminantium*, *Lactococcus* (gut)	Association is based on analysis of gut microbiota. HPV is detected in 40% of cases.	[23,57,83]
Breast fibroadenomas	*Staphylococcus aureus*, HPV (DNA detected in 38.9% of cases)	Not specified	*S. aureus* is considered a factor contributing to MED12 gene mutation.	[49,57,84]
Lactational mastitis	*Staphylococcus aureus* (main pathogen), coagulase-negative staphylococci, *Streptococcus*, *Pseudomonas aeruginosa*, *Escherichia coli*. Genera *Anaerofilum*, *Anaerotruncus* (gut)	Genus *Butyricimonas*, orders *Coriobacteriales*, *Pasteurellales*, *Verrucomicrobiales* (gut)	Lactational mastitis accounts for 33% of all breast diseases.	[21,23,57,85]
Non-lactational mastitis (including duct ectasia, periductal mastitis)	Family Prevotellaceae (gut). Genera *Ruminococcus*, *Coprococcus*, *Clostridium* (breast tissue).	Not specified	The breast tissue microbiota composition differs from that of healthy patients. May be associated with autoimmune reactions.	[23,29,85,86,87]
Granulomatous mastitis	*Corynebacterium kroppenstedtii* (key pathogen), genera *Pseudomonas*, *Brevundimonas*, *Stenotrophomonas*, *Acinetobacter*, fungi of the genus *Aspergillus*.	Not specified	Rare disease. Etiology is unknown; possible roles include dyshormonal changes and autoimmune reactions.	[21,29,85,87,88]
Periductal mastitis (duct changes)	Genera *Enterococcus*, *Streptococcus*, *Bacteroides*.	Not specified	Bacterial flora is detected in 50–62% of cases, often against the background of duct ectasia.	[85,89,90]
Purulent-septic changes (abscesses)	Coagulase-negative staphylococci, peptostreptococci, *Staphylococcus aureus* (including MRSA), *Corynebacterium*, *Pseudomonas aeruginosa*.	Not specified	Often occur as a complication of non-lactational mastitis. May be associated with impaired skin barrier function (e.g., dermatitis).	[21,29,85,91,92,93]
Fibrous capsular contracture (post-mammoplasty)	*Staphylococcus epidermidis* (most common), *Escherichia coli*, *Diaphorobacter nitroreducens*, *Cutibacterium acnes*, *Staphylococcus aureus*, *Staphylococcus* spp., *Pseudomonas aeruginosa*, *Sphingomonas paucimobilis*.	Not specified	Associated with bacterial colonization of the implant and biofilm formation.	[28,36,94,95,96,97,98,99]
Anaplastic large cell lymphoma (BIA-ALCL)	*Staphylococcus saprophyticus*, representatives of the genus *Ralstonia*.	Not specified	Associated with bacterial colonization and biofilms on breast implants.	[97,100,101]

## Data Availability

No new data were created or analyzed in this study. Data sharing is not applicable.

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
