# Peer review of "Mammary Gland Microbiota in Benign Breast Diseases"

_ijms, 2025, doi:10.3390/ijms26209951_

Round 1
Reviewer 1 Report
Comments and Suggestions for Authors
- Introduction
There are some issues that need improvement in terms of logical coherence, focus, concentration, and hierarchical structure, which makes the reading somewhat lengthy and scattered. It is suggested that the author reorganize the introduction slightly to create a more hierarchical and coherent "narrative arc".
- Logical connections and transitions can be stronger:
Problem: The transitions between paragraphs are somewhat abrupt. For instance, the first paragraph introduces the microbiome and biofilms, the second paragraph suddenly shifts to the definition and effects of dysbiosis, and the third paragraph jumps back to the research on the relationship between microorganisms and tumors. Although all the content is related, there are no smooth transitional sentences to guide the reader.
Suggestion: Introduce transitional sentences between paragraphs to make the writing flow more smoothly. For instance, after defining "Dysbiosis", one could write: "This dysbiosis has been linked to the pathogenesis of various tumors, prompting scientific research to explore the role of microorganisms in tumor formation at various anatomical sites..."
- Some sentences can be simplified to enhance the strength, and the precision of the use of terms should be ensured, especially for the terms. Please conduct a thorough check throughout the text.
Question: The first sentence at the beginning: Line 42 "Microorganisms play a crucial role as biological agents in both the external and internal environments of the human body". This sentence is a bit lengthy, and "biological agents" is not the most appropriate term in this context.
Suggestion: It can be directly changed to a more concise and powerful expression, such as: The human microbiome, comprising diverse microorganisms, is a critical factor in maintaining health and influencing disease within the host's internal environment and on its external surfaces."
- The focus can be further concentrated:
Question: The last part of the second paragraph mentions the role of microorganisms in tumor formation in various parts such as the stomach, intestines, liver, lungs, and skin. This example is somewhat broad. Since the focus of this article is on the breast, the topic can be narrowed down to the breast earlier.
Suggestion: This sentence can be modified to:“Scientific research has explored the role of microbes in tumor formation at various anatomical sites such as the stomach and intestines [10, 11], however, its role in breast pathology is only beginning to be understood.” This can lead readers to the core topic more quickly。
4.Strengthening the logical flow。
The above revision suggestions are for reference only.
- Pathways of mammary gland microbiota formation and mechanisms of its interaction with the host organism
- The abbreviations in the caption (such as MG-line 90, BBD), although explained in the caption, should also be defined for the first time in the main text to ensure that readers can understand the context without referring to the caption.
- 2 Is it necessary to place the content from lines 140 to 144 here?
- The caption for Figure 2 contains an unclear abbreviation ("Me1- monomethylation site"). This must be clarified or corrected.
- Suggestion: Replace "biological agents (2.3) "with the more neutral "microorganisms", "microbial constituents", or "biological components"
- Microbiota composition and diversity in BBD
6 Materials and Methods
This section is severely underdeveloped and does not meet the standards for a systematic review methodology.
Comments on the Quality of English LanguagePlease ensure the accuracy of the terminology used.
Author Response
Response to Reviewers
Manuscript ID: ijms-3915647
Title: Mammary Gland Microbiota in Benign Breast Diseases
We sincerely thank the reviewers for their thorough and constructive feedback on our manuscript. Their insightful comments have been invaluable in helping us improve the quality, clarity, and impact of our review. We have carefully considered all points raised and have revised the manuscript accordingly. Our point-by-point responses to the comments are detailed below.
Response to Reviewer 1
We thank Reviewer 1 for their thoughtful suggestions to enhance the logical flow, precision, and structure of our manuscript. All corrections made by Reviewer 1 are highlighted in yellow in the attached manuscript file.
Comment 1: Introduction: There are some issues that need improvement in terms of logical coherence, focus, concentration, and hierarchical structure... The transitions between paragraphs are somewhat abrupt.
Response 1: We agree with this assessment. We have thoroughly reorganized the Introduction to create a more coherent "narrative arc." Specifically, we have:
- Added smoother transitional sentences between paragraphs to guide the reader logically from general concepts (microbiome, dysbiosis) to the specific knowledge gap (breast microbiome in BBD).
- Incorporated the suggested transition, for example: "This dysbiosis has been linked to the pathogenesis of various health problems... Consequently, scientific research has explored the role of microbes in tumor formation at various anatomical sites... However, its role in breast pathology is only beginning to be understood."
Comment 2: Some sentences can be simplified to enhance the strength, and the precision of the use of terms should be ensured... The first sentence... is a bit lengthy, and "biological agents" is not the most appropriate term.
Response 2: We thank the reviewer for this suggestion. We have revised the opening sentence as recommended: "The human microbiome, comprising diverse microorganisms that exist predominantly as complex, structured communities known as biofilms, is a critical factor in maintaining health and influencing disease within the host's internal environment and on its external surfaces [1, 2]." We have conducted a thorough check throughout the text to simplify phrasing and ensure terminological precision.
Comment 3: The focus can be further concentrated... the example is somewhat broad. Since the focus of this article is on the breast, the topic can be narrowed down earlier.
Response 3: This is an excellent point. We have narrowed the focus in the relevant sentence to direct the reader to the core topic more efficiently. The text now reads: "Scientific research has explored the role of microbes in tumor formation at various anatomical sites such as the intestines, lung, skin and others [11, 12]. However, its role in breast pathology is only beginning to be understood."
Comment 4: *Pathways of mammary gland microbiota formation... The abbreviations in the caption (such as MG-line 90, BBD), although explained in the caption, should also be defined for the first time in the main text.*
Response 4: We have now ensured that all key abbreviations (MG, BBD) are defined upon their first appearance in the main body of the text, not only in the figure captions. We have also added a chapter of abbreviations at the end of the manuscript.
Comment 5: Is it necessary to place the content from lines 140 to 144 here?
Response 5: We believe this is appropriately placed in section 2.2, which describes the composition of the microbiota in unaltered tissue, as it provides a foundational explanation for the observed taxonomic profile. We have slightly rephrased this section for better integration.
Comment 6: *The caption for Figure 2 contains an unclear abbreviation ("Me1- monomethylation site"). This must be clarified or corrected.*
Response 6: We removed this from the signature and figure 1 itself.
Comment 7: Suggestion: Replace "biological agents (2.3) "with the more neutral "microorganisms", "microbial constituents", or "biological components"
Response 7: We have adopted this suggestion and replaced the term "biological agents" with "microorganisms" throughout the manuscript where appropriate.
Comment 8: Materials and Methods: This section is severely underdeveloped and does not meet the standards for a systematic review methodology.
Response 8: As this is a narrative review and not a systematic review or meta-analysis, we have revised the "Materials and Methods" section to more accurately reflect the nature of our literature search strategy. We have clarified the search databases, timeframe, keywords, and study selection process to enhance transparency, while explicitly stating its narrative scope.
Reviewer 2 Report
Comments and Suggestions for Authors
This manuscript, “Mammary Gland Microbiota in Benign Breast Diseases”, provides a comprehensive review of current knowledge on the origins, composition, and functional roles of the breast microbiota in benign breast conditions. The topic is timely and relevant, as most prior work has focused on breast cancer, while benign breast diseases (BBD) remain understudied despite their high prevalence and clinical significance. The authors successfully synthesize data on exogenous and endogenous microbiota pathways, disease-specific microbial signatures, and the limitations of existing research. However, in its current form, the paper is at times overly descriptive and would benefit from stronger critical analysis, clearer distinction of knowledge gaps, and greater emphasis on the translational implications for diagnosis and treatment.
- Abstract is quite dense; some sentences (lines 23–31) could be streamlined for clarity. The statement about “major limitations” (lines 31–34) would benefit from specifying examples rather than listing general categories.
- Rephrase abstract to emphasize the unique contribution of this review compared to prior work on cancer-focused microbiota.
- The introduction is lengthy and sometimes repetitive (lines 42–60 repeat general microbiome concepts already well known). Condense background on microbiome basics and instead expand on the knowledge gap in BBD microbiota research (lines 72–84).
- Some subsections (lines 93–119) are overly descriptive without critical synthesis of which pathways have the strongest evidence. Add a table or schematic ranking the strength of evidence for each pathway. Clarify in lines 153–167 how underexplored virome/mycobiome findings could affect BBD research.
- Lines 249–272: Comparisons of taxonomic profiles between malignant and benign conditions could be more critical—do these differences have diagnostic value, or are they merely descriptive?
- Lines 295–306: The discussion of fungi is brief; lacks integration with clinical implications.
- Strengthen interpretation of whether microbiota shifts are cause or effect in BBD. Highlight which taxa could serve as biomarkers (diagnostic/prognostic).
- Very limited data are cited at Breast Microbiota in Men; this reads more as speculation. Explicitly frame as a “research gap” rather than a developed conclusion.
- Clear listing of methodological and conceptual limitations (e.g., low microbial biomass, lack of standardized protocols). But some overlap with the Abstract’s limitation statement. Instead of a list, group limitations under categories (technical, clinical, conceptual) to improve readability.
- Search strategy and inclusion criteria are described. Lacks PRISMA-style flow diagram or systematic review rigor. Either clarify that this is a narrative review or strengthen transparency by specifying how many studies were included/excluded.
- Strong emphasis on future directions (multi-omics, mechanistic studies). Some conclusions repeat earlier discussion points (lines 526–532). Add a more clinical outlook such as how might microbiota research impact diagnosis or management of BBD in the next 5–10 years?
Author Response
Response to Reviewers
Manuscript ID: ijms-3915647
Title: Mammary Gland Microbiota in Benign Breast Diseases
We sincerely thank the reviewers for their thorough and constructive feedback on our manuscript. Their insightful comments have been invaluable in helping us improve the quality, clarity, and impact of our review. We have carefully considered all points raised and have revised the manuscript accordingly. Our point-by-point responses to the comments are detailed below.
Response to Reviewer 2
We thank Reviewer 2 for their positive assessment of our topic's relevance and for their insightful critiques aimed at strengthening the analytical depth and translational outlook of our review. All corrections made by Reviewer 2 are highlighted in green in the attached manuscript file.
Comment 1: Abstract is quite dense; some sentences (lines 23–31) could be streamlined for clarity. The statement about “major limitations”... would benefit from specifying examples... Rephrase abstract to emphasize the unique contribution.
Response 1: We have revised the Abstract to improve clarity and conciseness. We have also rephrased the opening to more sharply emphasize the review's unique focus on BBD, distinct from the cancer-centric literature.
Comment 2: Introduction is lengthy and sometimes repetitive... Condense background on microbiome basics and instead expand on the knowledge gap in BBD.
Response 2: We have condensed the general background information on the microbiome in the Introduction. The space saved has been used to expand upon the specific knowledge gaps and unresolved questions in BBD microbiota research, as suggested. We have also integrated the specific critical questions proposed by the reviewer into this section.
Comment 3: Some subsections (lines 93–119) are overly descriptive without critical synthesis... Add a table or schematic ranking the strength of evidence for each pathway.
Response 3: This is a valuable suggestion. While we have opted not to add a new table to avoid overcomplicating the figure set, we have significantly revised section 2.1 to include a critical synthesis of the evidence supporting each pathway. We now explicitly state the strength of evidence (e.g., "The evidence for this pathway is strong...", "Support for this pathway is moderate and indirect...", "Evidence for these pathways is currently limited...") for each exogenous route.
Comment 4: *Clarify in lines 153–167 how underexplored virome/mycobiome findings could affect BBD research.*
Response 4: We have expanded this section to discuss the potential implications of the virome and mycobiome for BBD research. We now explicitly state that their specific roles in BBD pathogenesis remain largely speculative and highlight this as a key area for future investigation.
Comment 5: Lines 249–272: Comparisons of taxonomic profiles... could be more critical—do these differences have diagnostic value, or are they merely descriptive?
Response 5: We have strengthened this section by adding critical analysis. We now explicitly question the diagnostic utility of these findings given current methodological limitations and pose the critical question of whether these signatures are drivers of pathogenesis or secondary consequences. We also highlight the potential for future biomarker development as an unvalidated but promising area.
Comment 6: Lines 295–306: The discussion of fungi is brief; lacks integration with clinical implications.
Response 6: We have substantially expanded the discussion on fungi (mycobiome). We now draw parallels with findings in breast cancer and propose plausible mechanisms (e.g., sustained inflammation in granulomatous mastitis) and potential clinical implications (e.g., antifungal therapies) for BBD, while clearly framing these as hypotheses awaiting validation.
Comment 7: Strengthen interpretation of whether microbiota shifts are cause or effect in BBD. Highlight which taxa could serve as biomarkers.
Response 7: We have added text throughout the review, particularly in the "Microbiota composition and diversity in BBD" section and the Conclusions, to more critically discuss the causality question. We now more explicitly highlight specific taxa (e.g., Corynebacterium kroppenstedtii, gut Clostridium/Faecalibacterium) as compelling biomarker candidates and discuss the contextual evidence that might suggest a causative role.
Comment 8: Very limited data are cited at Breast Microbiota in Men; this reads more as speculation. Explicitly frame as a “research gap”.
Response 8: We agree entirely. We have reframed the entire section on "Breast Microbiota in Men" to explicitly present it as a significant research gap. We highlight the paucity of data, describe the existing preliminary findings as such, and conclude by stating the necessity for targeted future investigation.
Comment 9: Clear listing of methodological and conceptual limitations... Instead of a list, group limitations under categories (technical, clinical, conceptual) to improve readability.
Response 9: This is an excellent organizational suggestion. We have completely restructured the "Limitations" section (now section 5) into three clear sub-sections: 5.1. Methodological and Analytical Limitations, 5.2. Clinical and Cohort-related Limitations, and 5.3. Conceptual and Interpretative Limitations. This improves readability and provides a more structured critique of the field.
Comment 10: Materials and Methods... Lacks PRISMA-style flow diagram or systematic review rigor. Either clarify that this is a narrative review or strengthen transparency...
Response 10: We have revised the "Materials and Methods" section to clarify that this is a narrative review. We have also enhanced its transparency by providing more detail on the search strategy, including the number of records identified, screened, and ultimately included, without presenting it as a systematic review. We have added the sentence: "While not a systematic review, this approach aimed to comprehensively cover the current state of knowledge..."
Comment 11: Strong emphasis on future directions... Add a more clinical outlook such as how might microbiota research impact diagnosis or management of BBD in the next 5–10 years?
Response 11: We thank the reviewer for this forward-looking suggestion. We have expanded the Conclusions section to include a more concrete clinical outlook. We now discuss potential translational applications in the near future, such as the use of microbial signatures for differential diagnosis of BBD subtypes, risk stratification for malignant transformation, and the development of novel microbiota-modulating therapies.
We believe that the revisions we have made in response to the reviewers' comments have significantly strengthened the manuscript. We are grateful for the opportunity to improve our work and hope that the revised version is now suitable for publication.
Round 2
Reviewer 2 Report
Comments and Suggestions for Authors
The revised manuscript fully addresses previous concerns, and I have no further comments.